# Decision making biases in the allied health professions: A systematic scoping review

**Rebecca Featherston**[1], **Laura E. Downie**[2], **Adam P. Vogel**[3,4,5,6], **Karyn L. Galvin**[6]*

**1** Department of Social Work, School of Primary and Allied Health Care, Faculty of Medicine Nursing & Health Sciences, Monash University, Melbourne, Victoria, Australia, **2** Department of Optometry and Vision Sciences, Faculty of Medicine, Dentistry and Health Sciences, The University of Melbourne, Melbourne, Victoria, Australia, **3** Centre for Neuroscience of Speech, Melbourne School of Health Sciences, The University of Melbourne, Melbourne, Victoria, Australia, **4** Department of Neurodegeneration, Hertie Institute for Clinical Brain Research, University of Tübingen, Tübingen, Germany, **5** Redenlab, Melbourne, Victoria, Australia, **6** Department of Audiology & Speech Pathology, Melbourne School of Health Sciences, The University of Melbourne, Melbourne, Victoria, Australia

* kgalvin@unimelb.edu.au

**Data Availability Statement:** All relevant data are within the manuscript and its Supporting Information files.

**Funding:** This work was supported in part by a Mid-Career Researcher Grant from The University

## Abstract

### Objectives

Cognitive and other biases can influence the quality of healthcare decision making. While substantial research has explored how biases can lead to diagnostic or other errors in medicine, fewer studies have examined how they impact the decision making of other healthcare professionals. This scoping review aimed to identify and synthesise a broad range of research investigating whether decisions made by allied health professionals are influenced by cognitive, affective or other biases.

### Materials and methods

A systematic literature search was conducted in five electronic databases. Title, abstract and full text screening was undertaken in duplicate, using prespecified eligibility criteria designed to identify studies attempting to demonstrate the presence of bias when allied healthcare professionals make decisions. A narrative synthesis was undertaken, focussing on the type of allied health profession, type of decision, and type of bias reported within the included studies.

### Results

The search strategy identified 149 studies. Of these, 119 studies came from the field of psychology, with substantially fewer from social work, physical and occupational therapy, speech pathology, audiology and genetic counselling. Diagnostic and assessment decisions were the most common decision types, with fewer studies assessing treatment, prognostic or other clinical decisions. Studies investigated the presence of over 30 cognitive, affective and other decision making biases, including stereotyping biases, anchoring, and confirmation bias. Overall, 77% of the studies reported at least one outcome that represented the presence of a bias.

of Melbourne's School of Health Sciences (KG, LED). This funder provided support in the form of salaries for authors [RJF] and research materials, but did not have any additional role in the study design, data collection and analysis, decision to publish, or preparation of the manuscript. The specific roles of all authors are articulated in the 'author contributions' section. A.P.V. received salaried support from the National Health and Medical Research Council, Australia (Dementia Fellowship ID 1135683), and also has a commercial affiliation with Redenlab. Neither of these funders played a role in this study.

**Competing interests:** A.P.V. has a commercial affiliation with Redenlab, however this did not play a role in this study. This does not alter our adherence to PLOS ONE policies on sharing data and materials.

## Conclusion

This scoping review provides an overview of studies investigating whether decisions made by allied health professionals are influenced by cognitive, affective or other biases. Biases have the potential to seriously impact the quality, consistency and accuracy of decision making in allied health practice. The findings highlight a need for further research particularly in professional disciplines outside of psychology, using methods that reflect real life healthcare decision making.

## Introduction

Quality health service delivery is dependent on good decision making. An important component of this is the process of clinical reasoning that leads to a decision being made by a health professional. Human cognitive resources are limited, affecting how information is processed, stored and retrieved [1]. Both intrinsic cognitive and extrinsic environmental or systemic factors can influence the way information is used to make decisions. In the context of decisions made in healthcare, any issues associated with the process of clinical reasoning can have serious ramifications for patients, clients and clinicians. Our cognitive reasoning processes, either inherent ('hard wired') or learned, can lead to systematic deviations from the rules of logic and probability, concepts that are largely considered to be the basis of rational thinking [2,3]. These deviations are often referred to as decision making biases. The classification and conceptualisation of biases is complex [3–6]. Nevertheless, many biases have been described in the literature, and many of these, including cognitive, stereotyping, attributional and affective biases, have been shown to influence decision outcomes [5]. There is also evidence suggesting that these reasoning errors can influence decisions in numerous professional scenarios, including in the delivery of health and social services [7–10].

Reasoning biases challenge the assumption that humans make logically correct decisions when they are provided with sufficient information. The introduction of the term *cognitive bias* was particularly influential [11]. This concept influenced theory across multiple disciplines including the social sciences, psychology and economics [2]. Cognitive shortcuts, known as "heuristics", decrease the cognitive burden of decision making by minimising the amount of time and information required to make a decision [2,12,13]. For example, the *availability heuristic* is a heuristic that describes a cognitive process whereby the information most readily retrieved from a person's mind is used to make a decision [5,14]. Heuristics can be efficient, often resulting in accurate decision outcomes, however they are also vulnerable to systematic reasoning errors, or cognitive biases [5,11,12]. For instance, the availability heuristic can produce a number of biases because the most readily retrievable information may not be the most accurate or appropriate information to support the decision being made. It may, for example, simply reflect how recently the information was obtained (recency bias/effect) [5,11]. Since Tversky and Kahnemann's pivotal work in the early 1970s, over one hundred cognitive biases have been named and described, and many have been tested for their effect on decision making in diverse scenarios [5,14,15].

Cognitive biases arising from heuristics-based thinking are not the only type of reasoning errors that can influence human decision making. Other biases can result from the complex neurological interplay between cognition and emotion, as well as their interaction with the external social context. Affective biases can occur when a person's emotional state influences decision outcomes. For instance, anxious and depressed people are more likely to make negative judgements and to interpret the same stimuli negatively than are people who are not

depressed or anxious [16–19]. The fundamental attribution error (correspondence or attribution effect) describes the systematic tendency to overestimate personality or dispositional causes and to underestimate situational causes when making judgements about the behaviour of others [15,20–22]. Stereotyping biases result from a set of cognitive over-generalizations (e.g., beliefs, expectations) about the qualities and characteristics of members of a group or social category that are not revised even following an encounter with an individual with qualities that are not congruent with the stereotype [23]. Age biases, sex or gender biases, and racial biases are some examples. The thought processes that result in biases are complex. For instance, the cognitive, affective and social processes that regulate stereotyping likely vary depending on the type of bias. Implicit stereotypes, those that are unconsciously applied to judgements about others, have been shown to involve neural structures relating to semantic memory, while implicit racial biases are more closely associated with affective memory systems [24]. Here, we use the term 'bias' or 'biases' to represent any type of cognitive, affective or other bias that may systematically alter a decision outcome.

Biases have the potential to influence the quality of healthcare decisions. There is a considerable body of evidence demonstrating that medical practitioners are vulnerable to a range of biases when making decisions, particularly when making diagnostic and treatment decisions [7,10,25]. A review by Saposnik et al [10]. compiled studies that identified cognitive biases contributing to medical decision making errors made by physicians, which included anchoring and framing effects, availability bias, satisficing and confirmation biases, overconfidence and risk tolerance. All of the included studies found at least one cognitive bias or personality trait that affected physician decision making. A critical review by Blumenthal-Barby and Krieger [5] identified 213 studies reporting on 19 different cognitive biases and heuristics affecting clinician and consumer medical decision making. They found that 68% of the included studies detected the presence of a bias. There is also evidence that implicit biases affect medical decisions [26,27]. A recent systematic review showed that physicians and nurses can be vulnerable to several implicit biases when interacting with their patients, including biases related to race, ethnicity, gender, age, socioeconomic status, weight, disability, disease-type, drug use, and mental health [26].

Given the pervasive influence of decision making biases in the general population, and the considerable evidence suggesting that biases can have a negative influence on medical decision making, it is probable that similar biases impact the decision making of allied health professionals. By definition, allied health professionals are health professionals who are not part of the medical, dental or nursing professions. They are university-qualified professionals with specialised expertise in preventing, diagnosing and treating a range of conditions and illnesses. They make important decisions that impact the health and quality of life of their clients/ patients and so investigating their decision making is important. There are obvious parallels in the types of decisions and decision processes undertaken by allied health professionals and medical professionals, and therefore biases likely play an equally significant role. There are many scenarios across the spectrum of allied health service delivery that may potentially be vulnerable to biases. For instance: the amount or type of information in a referral letter may alter the type of assessment undertaken or service provided (anchoring bias); a practitioner may be less inclined to alter a prior assessment or diagnosis, and/or change the services currently provided, if key patient/client information is received at a later stage of service delivery (diagnostic momentum; confirmation bias; order effect); the experience of a recent case may influence how a later case is assessed or diagnosed (availability bias; recency effect), or; the physical appearance of a patient or client may alter the assessment or diagnosis, or services provided (stereotyping bias).

Perhaps surprisingly, seemingly few studies have investigated bias in healthcare professions other than medicine, and there are few systematic reviews synthesising relevant evidence. Hall et al. [28] conducted a systematic review to identify studies that assessed whether implicit racial bias was demonstrated among healthcare professionals, including medical, nursing and allied health professionals, and whether this in turn altered health outcomes. A majority of the 15 identified studies reported low to medium levels of bias, and implicit racial bias was significantly related to patient–provider interactions, treatment decisions, treatment adherence, and patient health outcomes [28]. Notably, while this review considered a broad range of health professions, none of the identified studies included allied health professionals. Several older reviews have explored the role of racial, socioeconomic status, sex and gender biases in clinical psychotherapeutic evaluations, and also discuss the methodological challenges of confirming the presence of these biases [29–32]. However, there is no existing review that explicitly searched for literature that included a test to determine whether decisions made by allied health professionals were influenced by cognitive, affective or other biases.

This study's primary objective was to undertake a scoping review to identify studies that tested whether cognitive, affective or other biases influenced decisions made by allied healthcare professionals. Scoping reviews are useful for mapping a broad range of evidence to determine what research has been conducted to date [33,34]. By identifying a wide range of evidence, this review aimed to ascertain; 1) In which allied health professions have the effect(s) of bias on professional decision making being considered? 2) In which health settings have biases been tested? 3) What biases have been considered to potentially influence allied health professionals' decision making? 4) What allied health decisions have been tested with respect to the influence of bias?

## Materials and methods

The methods for this scoping review were guided by the PRISMA Extension for Scoping Reviews (PRISMA-ScR) [34]. A completed PRISMA-ScR checklist has been provided in the supporting information (see **S1 File**). All review methods were determined *a priori* and written into a protocol, before commencing literature searches and screening. While registration was attempted, PROSPERO was not accepting scoping review protocols at that time. The original protocol can be provided on request.

### Eligibility criteria

The following eligibility criteria were used to determine whether to include or exclude studies identified by the electronic literature searches.

**Participants.**   Decision making participants included any graduate student or fully trained allied healthcare professional. There are several existing definitions for the term 'allied health', and the professions that fit under this umbrella term can vary substantially between countries. Here, we have used the definition ascribed by Allied Health Professions Australia (AHPA), alongside the list of professions provided at https://ahpa.com.au/what-is-allied-health/ (see **S2 File**). Studies were also considered for inclusion when the participants fit the AHPA definition, whether or not the profession was included on the AHPA list. This definition was selected because it restricts the term 'allied health' to professionals who work directly with clients/patients, which was appropriate for the scope of this review. Studies that included undergraduate student samples were not included. Studies that reported medical or nursing professionals as the primary participants were only included if there were also allied healthcare professional participants. Studies that focussed on consumer decision making or explored decision making and bias within other professions were excluded.

**Settings.**   Eligible service settings included hospitals, clinics, community centres, schools, private homes, universities or any other settings where allied health services can be administered or taught by a provider. Studies that used hypothetical or simulated scenarios, vignettes or surveys were also included, provided that the scenario, vignette or survey directly related to a decision that would be made by an allied healthcare professional.

**Decision types.**   We defined decision making as any decision made within the context of allied health service delivery and, more particularly, decisions made within a clinical context by allied health professionals. Studies reporting on decisions relating to screening, assessment, diagnosis, prognosis and treatment were all eligible, as were studies reporting other decision types not identified here.

**Cognitive, affective and other biases.**   Studies that tested for the presence of any cognitive bias defined in the literature (e.g., confirmation bias, hindsight bias, order effects) were eligible. Studies that specified a cognitive bias or error, or used the terms 'cognitive bias/es' or 'cognitive error/s', as well as studies that used more general terms, such as 'clinical reasoning errors' or 'errors in clinical reasoning', were eligible for inclusion. We also included studies that investigated the presence and influence of decision making biases produced by affective (emotional) states, attributional effects and stereotyping (i.e., racial bias, gender bias, age bias). We use the term 'bias' or 'biases' to represent any of type of cognitive, affective or other bias that may systematically alter a decision outcome.

**Outcome measures.**   The review aimed to identify studies involving outcomes that represented the presence or absence of biases arising from a decision made by allied healthcare professionals. We focused specifically on outcomes that aimed to detect changes in proximal decision outcomes that were designed and used to represent a specified cognitive, affective or other bias. For example, a diagnostic case study may have two versions specifically designed to test for a cognitive bias. That is, the way information is presented within the two versions, to two separate comparison groups of participants, may differ in a specific way (such as the ordering, framing or anchoring of the information). Where there is a difference in the diagnostic decision made by the two groups, this would reflect the presence of a cognitive bias (such as an order effect, framing effect or anchoring bias). Different kinds of outcomes may reflect the presence or absence of a bias and these were all eligible for inclusion.

**Study designs.**   Studies were deemed eligible if the study design included comparators that allowed for an assessment of the presence or absence of a bias influencing the decision making of an allied health professional. This included, but was not limited to, randomised controlled trials (RCTs), non-randomised controlled trials (nRCTs), and pre-post designs with a control group. Study designs involving any comparator/s were included. Studies that did not include a comparison condition were not included.

## Information sources

An electronic search strategy was designed to identify studies that tested for the presence of bias in the decision making of healthcare professionals across multiple allied health disciplines. Keywords relating to "decision making", "professional discipline", and "bias", were used to search the following electronic databases in December, 2018 (see **S3 File**): (1) *Ovid MEDLINE (R)*, *Ovid MEDLINE(R) In-Process & Other Non-Indexed Citations*, *Ovid MEDLINE(R) Daily and Ovid OLDMEDLINE(R) (1946 to present)*; (2) *Cumulative Index to Nursing and Allied Health Literature (CINAHL) via Ebsco (1980 to present);* (3) *PsycINFO (Ovid*, and; (4) *SCOPUS (Elsevier)*. Text word searches were mapped verbatim into each database, excepting adjustments made for database specific syntax. Grey literature was searched to identify studies not

indexed in the databases listed above, using *Proquest Dissertation and Theses*. The reference lists of all included studies were also hand searched for additional eligible primary studies.

## Data management and software

Reference management software *EndNote X7* (Clarivate Analytics, 2017) was used to compile all titles and abstracts derived from the initial searches, and duplicates were removed. Multiple *EndNote X7* files were used to manage retrieved records, screen studies, and to identify discrepancies across reviewers.

## Study selection

Prior to study selection, all review authors underwent training to ensure a comparable understanding of the purpose of the review and the eligibility criteria. Titles and abstracts retrieved from the electronic searches were screened to exclude publications that did not meet the eligibility criteria. This stage of the screening process was highly inclusive, and a full text review was undertaken when the information provided in the titles and abstracts was unclear or insufficient. For the full text screening, each of the studies was assessed independently (blind) by two review authors. All screening discrepancies were discussed, with any outstanding conflicts resolved by a third review author. Four review authors in total, contributed to the screening process. A flow diagram, outlining the study selection process, was completed following the methods outlined by the Preferred Reporting Items for Systematic Reviews and Meta-Analyses (PRISMA) statement [35].

## Data collection

A data extraction form was developed *a priori*. The following data were extracted: study characteristics (e.g., authors, year published, country), bias type, decision type, participants (including sample information, e.g., sample size, allied health profession), comparators, outcomes representing the presence/absence of bias and study design. A single review author extracted all data, and a second review author independently checked the extracted data for potential errors. Any discrepancies were discussed and agreement was reached.

# Results

## Study selection and study characteristics

**Fig 1** outlines the electronic search and screening process, which resulted in 149 studies being assessed as meeting the eligibility criteria from a total 9376 citations identified from the search strategy. Of the 149 studies, 95 were published in peer-reviewed journals. Two of these were journal publications of a study originally reported in a dissertation [36,37], and these have been excluded from the following results. The sample and reported results of one other study differed between the dissertation and journal publication versions [38,39]. A summary of the included studies and a reference list has been provided in the supporting information (**S4 File**). Publication years ranged from 1956 to 2018, with few included studies published between 1956 and 1970 (n = 5), followed by a fairly consistent number of studies being published each decade from 1970 onwards (range: 20 to 33 studies per decade). Most studies were undertaken in the United States (n = 129), with the remainder undertaken in Canada (n = 4), Germany (n = 3), United Kingdom (n = 3), Israel (n = 3), Australia (n = 2), Netherlands (n = 1) and France (n = 1). One study originated from both the United States and Canada (n = 1).

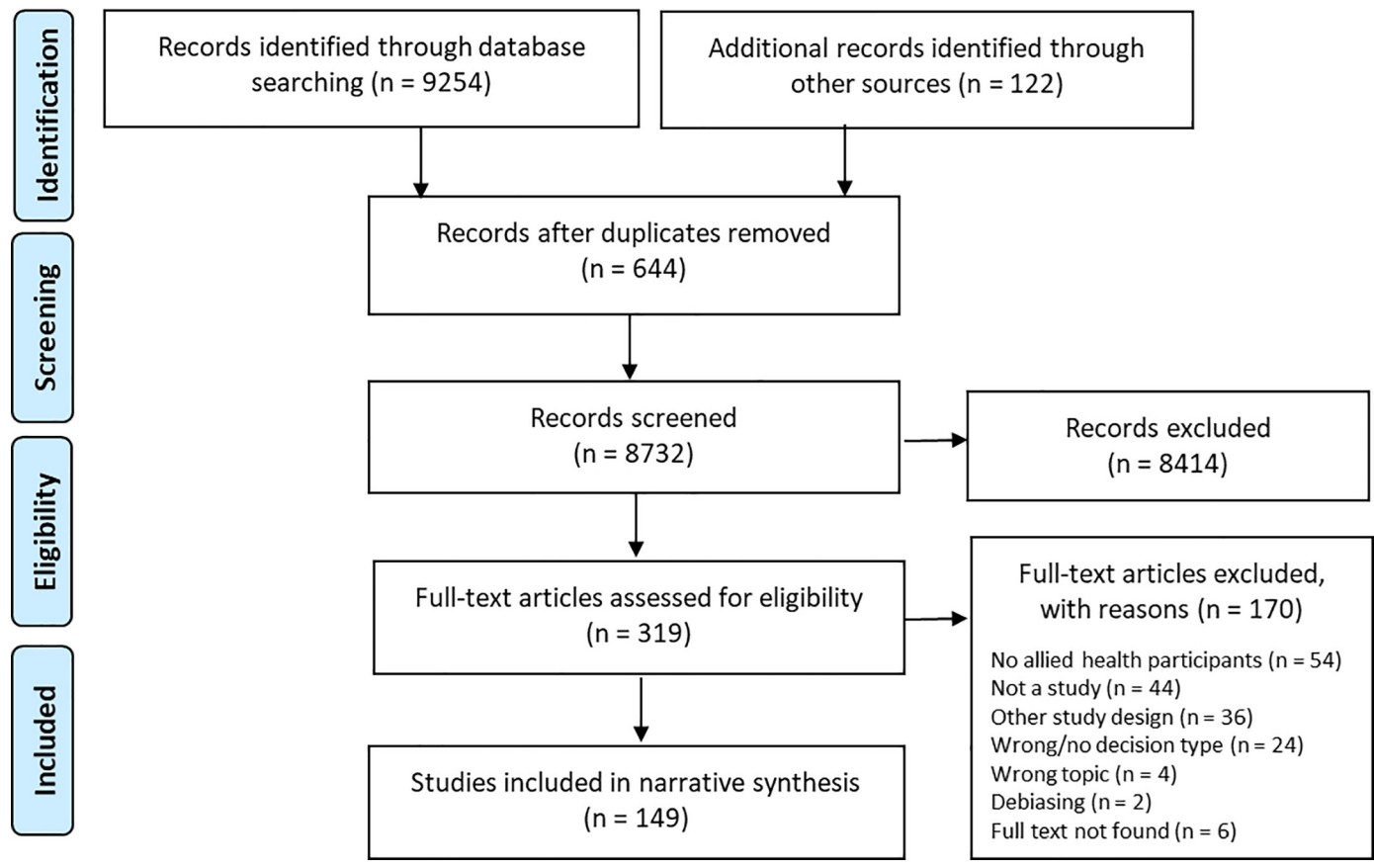

**Fig 1. PRISMA flow diagram reporting the electronic database search, screening and study selection process.**

## Allied health professionals

See **Fig 2**; **S4 File**. Most studies (82%) included clinical, counselling or school psychology participants, either fully qualified professionals or graduate students (n = 121). Masters-level students and fully qualified social workers were included as participants in 32 studies, but 16 of these were mixed and also included participants who were psychology professionals or graduate students (see **S4 File**). Speech pathologists and/or speech-language pathologists were the primary participants in 11 studies. The decision making of physical therapists and audiologists' was assessed in two studies respectively, and occupational therapists and genetic counsellors were included in a single study each. There was some overlap; one study included rehabilitation professionals, a term which encompassed psychologists, physical therapists, occupational therapists and speech pathologists [40], and one study that included audiologists also included speech pathologists [41]. Several allied health professions, including optometry, osteopathy, dietetics and sonography, were not represented by any study.

## Settings

See **S4 File**. Almost all included studies (n = 148) used hypothetical scenarios. These included clinical vignettes or case analogues (either written, audio or videotaped) where the bias of interest was demonstrated by experimentally manipulating some part of the information presented to participants (e.g., a characteristic of the case character, referral information, information order) across the test conditions. A single study [42] used real life data. This study

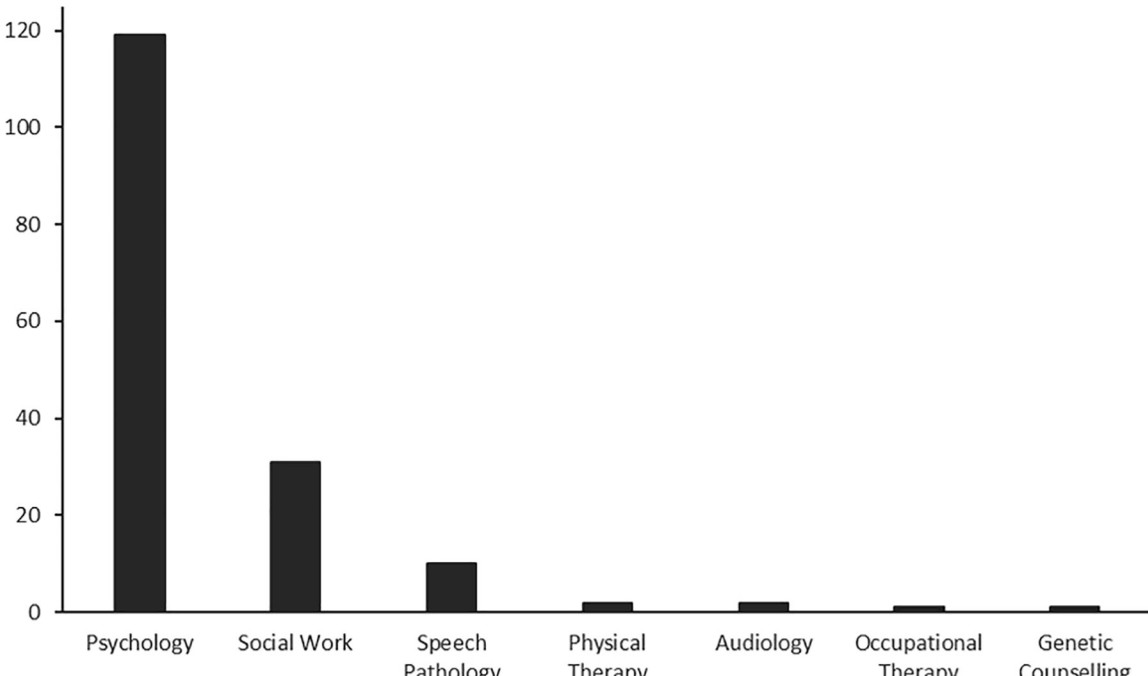

**Fig 2. Number of studies that included participants from each of the allied health professions.**

categorised pairs of clients and therapists by race to detect bias in therapists' assessment of clients and their treatment outcomes.

## Decision types

See **Fig 3**; **S4 File**. Decision types reported within the included studies were grouped into seven broad categories: assessment (n = 86); diagnosis (n = 68), treatment (n = 38), prognosis (n = 34), child placement (n = 6), school placement (n = 4 studies), and genetic likelihoods (n = 1). Eighty-two studies assessed the impact of a bias on a single decision type (represented by one or more outcome measures), with the remainder assessing more than one decision type. 'Assessment' was a broad category and included any kind of clinical assessment of a patient/client. 'Child placement' referred to the decision to place a child or young person into out-of-home care, and 'school placement' included the decision to refer a student to a non-mainstream school or alternate learning program. We grouped decision types based on the description provided by the study authors, therefore there is likely overlap across diagnostic, assessment, and prognostic decision groups depending on how the authors described their outcome measures.

## Cognitive, affective or other biases

See **Fig 4**; **S4 File**. A large proportion of the included studies represented biases related to stereotyping, including those associated with sex/gender or sex roles (n = 40), racial/ethnic status (n = 35), socioeconomic/class status (n = 26), age (n = 8), sexual orientation (n = 5), health status (n = 3), mental health (n = 2), weight (n = 1), speech intelligibility (n = 1), drug use (n = 1), and type of child maltreatment (in the context of child protection reports) (n = 1). There were 27 other biases reported across the included studies. Anchoring (n = 11), confirmation bias (n = 12), diagnostic overshadowing (n = 5) and labelling bias (n = 5) were the most common

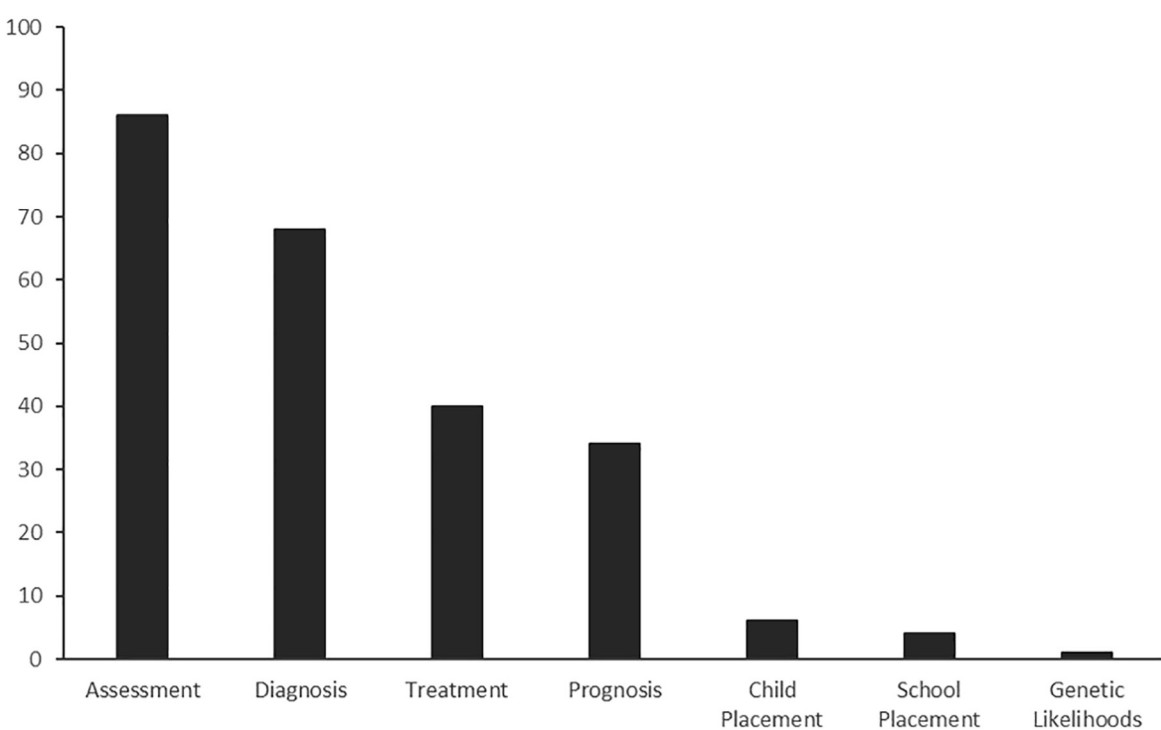

**Fig 3. Number of studies reporting each broad decision type.**

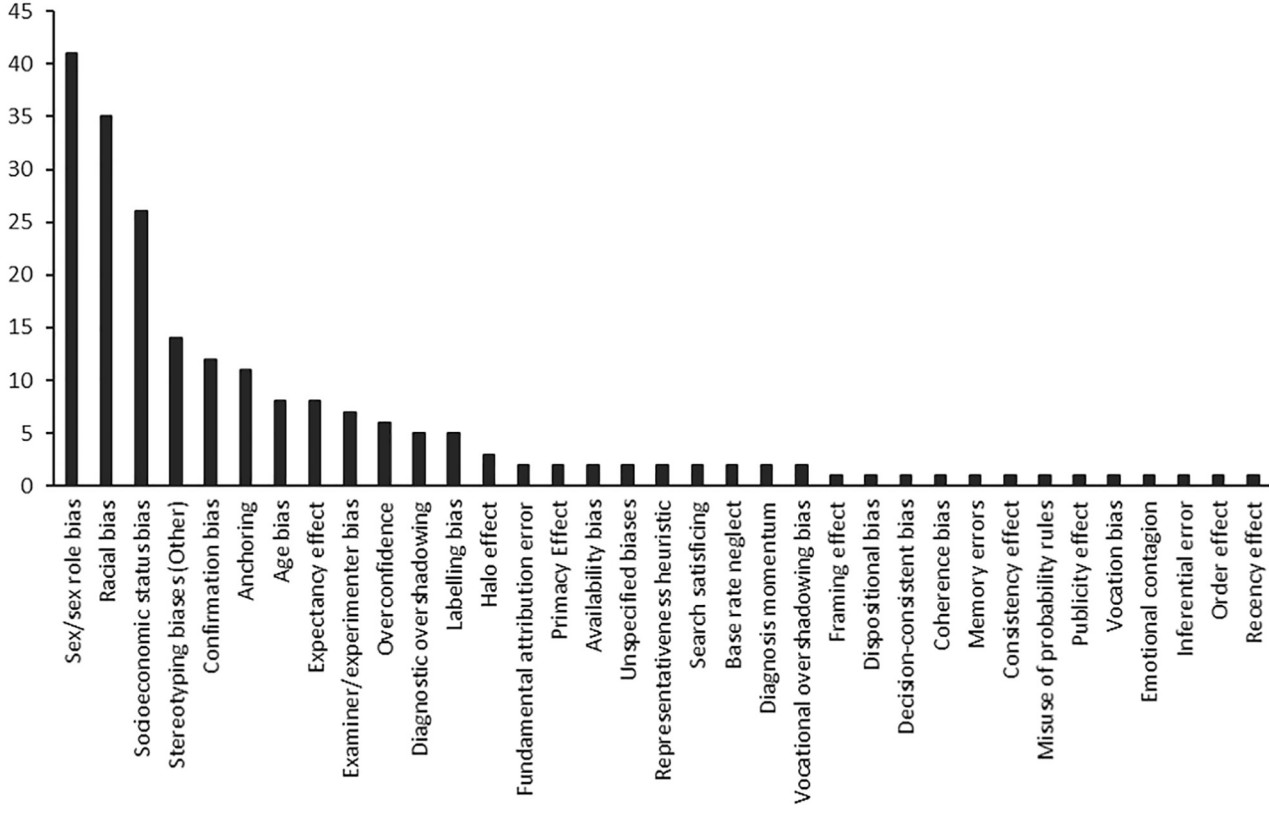

**Fig 4. Number of studies reporting each decision making bias.**

cognitive biases tested across the studies. Some studies, particularly those published pre-1980, used general bias terms, such as examiner/experimenter bias (n = 7) and expectancy effect (n = 8). Of the total 149 included studies, 77% reported a result on at least one outcome that represented the presence of a bias (see **S3 File**).

## Discussion

We identified 149 studies that tested whether a cognitive, affective or other bias influenced a clinical decision being made by an allied health professional. Most studies included participants trained in clinical, counselling or school psychology, with fewer representing other allied health professions. Almost all (148 studies) used methods that involved hypothetical scenarios to determine how a bias influenced a decision outcome. Racial/ethnic, socioeconomic, age and other stereotype related biases were the most common biases assessed across the studies. Twenty-seven other biases were assessed, with anchoring and confirmation bias the most frequently tested cognitive biases. Decisions relating to diagnosis and/or assessment were more likely to be evaluated than treatment choice, prognosis or any other clinical decision type.

Studies involving graduate psychology students or professionals were more common than those involving any other allied health profession. This is perhaps not surprising, given that the broader field of psychology is focussed on the scientific study of the brain and behaviour. As such, much of the testing of the impact of cognitive biases on decision making in all contexts has been conducted within the field of psychology, or by psychologists examining human behaviour in fields such as economics. Furthermore, in comparison to psychology, the other allied health professions tend to be smaller, more recently established, and with a greater focus on clinical care relative to research. For example, there are 106,500 licensed (clinical, counselling or school) psychologists in the United States, but only 16,000 licensed audiologists. Despite the differences across professions, there are obvious similarities in the decision making processes required across healthcare professions, making it probable that bias would play an equally important role in decision outcomes in all allied health fields, as has been shown to be the case in medicine, nursing and psychology [7,10]. Allied health professionals work with similarly diverse patient/client populations in a comparable range of healthcare settings, and also experience analogous resource constraints that may increase the likelihood of decision making biases. While not all allied health professions have clear diagnostic criteria, formal assessment of patients/clients is universal. Likewise, most allied health professionals make treatment recommendations and also provide prognoses. Therefore, a lack of relevance is not likely to explain the scarcity of research into how biases influence decision making of allied health professionals.

All except one study [42] used hypothetical clinical scenarios or vignettes to assess differential decision outcomes, modifying key bias-relevant information in an otherwise identical case presentation. This methodology has also been frequently used to explore bias in medical decision making [7]. While hypothetical scenarios provide a mechanism for experimental control over variables of interest, they have several limitations, particularly when extrapolating findings to decision making in real life contexts. For instance, resource or time limitations, modes of communication, or assessment measures may differ between real life contexts and hypothetical scenarios. Studies assessing decision making in real life contexts will be integral to further developing this area of research. That said, assessing clinical decisions using carefully constructed hypothetical scenarios, developed to answer specific questions, can be highly generalisable to real life behaviour [43]. Using vignette methodologies is less resource intensive, poses fewer ethical constraints and can overcome scientific limitations that exist when assessing decision making in real life contexts. These approaches may be more practical, particularly in

allied health professions where little or no research exists, and where initial exploratory studies will likely need to focus on identifying which biases and decision types will provide worthwhile avenues for further research.

In relation to the type of biases examined, most biases were reported within a single study only. Stereotyping biases, anchoring, and confirmation bias were examined in the greatest number of studies. While it is possible that this reflects the applicability of certain biases to allied health decision making, it more likely reflects a historical interest in these biases. Given the large number of biases recognised in the literature [5,14,15], there is extensive scope for further research. However, future studies should closely examine whether the methods used to detect systematic deviations in decision outcomes are sensitive enough to detect the bias under investigation. For instance, stereotyping biases may result in suboptimal decision outcomes due to the stereotype not accurately representing the individual, group or population it represents. Very few studies investigated whether characteristics associated with the stereotype in question accurately represented the population of interest [44,45]. Ford and Widiger [44] assessed whether sex biases in the diagnosis of antisocial and histrionic personality disorders could be explained by base rate (prevalence) differences between the sexes. They found that differential diagnoses based on sex could not be explained by underlying base rates, even for less ambiguous case scenarios. Another potential issue for future work is the lack of consistency when defining biases of interest. In some cases, the experimental design across multiple studies was highly comparable, but the studies themselves aimed to test different biases. For instance, expectancy effects [46–48], confirmation bias [49–51], and anchoring [52,53]. Or, a general rather than specific bias term was used, making it difficult to determine what specific mechanism was predicted to drive the differences in the decision outcomes. For example, 'examiner/experimenter bias' or 'expectancy bias' are general terms used in some publications prior to 1980.

Diagnostic and assessment decisions were the most studied decision types, followed by treatment, prognosis, or other decisions. Assessing the existence of bias can depend on the ability to define a clear decision outcome. This can be more straightforward for some health disciplines. For example, clear diagnostic guidelines exist for mental health disorders, allowing for decision outcomes to be assessed against a professional standard. This is not always the case for clinical decisions made in other allied health professions, particularly where decision outcomes rely more heavily on clinical expertise and judgement, than on following standardised criteria. This may be further complicated by the reality that many decisions are made by a group of professionals, rather than by an individual. This poses challenges to making predictions for how a bias might influence both the decision making process and a decision outcome, as well as identifying whether a decision made under a bias condition is more or less accurate, or appropriate, for the scenario in question. A lack of clarity around defining decision outcomes may also lead to additional issues when comparing results across studies. For instance, within this review, there is likely overlap between studies that defined a decision as 'assessment' and those using the term 'diagnosis' because some assessment tools included diagnostic components, but this was not always reported within the studies. This reduces the clarity around what actual clinical decision is impacted by a bias under investigation.

Of the total 149 included studies, 77% reported at least one outcome that represented the presence of a bias. It is an assumption made across most of the studies that the presence of bias has a negative impact on the decision making process and consequently the decision outcome for a client/patient. It raises serious concerns that a different clinical decision can be made as a result of a single piece of seemingly irrelevant information (e.g., the ethnic background of the client/patient) which has been included within otherwise identical case scenarios, or as a result of important case information being presented in a different context (e.g., in a different order).

That said, most studies did not address whether the decision made was more or less accurate, or whether the decision making process included more errors when a bias was detected. One study [38,39] showed that four separate biases were associated with both diagnostic accuracy and decision making error status in the diagnosis of paediatric bipolar disorder. This study also showed that inaccurate, and somewhat accurate, diagnostic decisions were significantly associated with different treatment and clinical recommendations. It will be important for future studies to be able to show that biases result in suboptimal clinical decision making and decision outcomes, leading to inappropriate treatment and/or service provision which results in worse outcomes for clients/patients.

Overall, the body of evidence presented here does suggest that the presence of bias produces inconsistencies in decision making and decision outcomes, and this is a considerable concern in its own right. It indicates that a patient/client may be assessed, diagnosed, and/or receive a different treatment or service as a result of the presence or absence of information that should not have played a role in the decision making process. It is therefore not an unreasonable assumption that cognitive, affective or other biases are likely to have a negative overall effect on the quality of clinical decision making.

This review has some limitations. While many known biases and allied health fields were included in the electronic search strategy, not all were included, so some studies may not have been captured. The aim of this scoping review was to gain a broad landscape of the existing literature exploring biases in allied health professionals' decision making. Therefore, a quality (risk of bias) assessment of the included studies was not undertaken and, as such, we can say very little about whether the studies are likely to have accurately detected the bias under examination. For this reason, we have not reported the specific outcomes of the studies.

## Conclusion

Biases have the potential to seriously impact the quality, consistency and accuracy of decision making in allied health practice. This scoping review provides an overview of studies investigating whether decisions made by allied health professionals are influenced by cognitive, affective or other biases. The findings highlight a need for further research in professional disciplines outside of psychology, using methods that better reflect real-life healthcare decision making. There is considerable opportunity for future research to explore the types of biases that influence key decisions made by allied health professionals across the spectrum of service delivery, and significant potential for the development of interventions to mitigate their negative impact in clinical practice [8,54,55].

## Supporting information

**S1 File. Preferred Reporting Items for Systematic reviews and meta-analyses extension for scoping reviews (PRISMA-ScR) checklist.**
(PDF)

**S2 File. Definition for allied health professionals included in the review.**
(DOCX)

**S3 File. Example database search strategy.**
(DOCX)

**S4 File. Summary of the included studies (n = 149).**
(DOC)

**S5 File. References (Included Studies).**
(DOC)

## Acknowledgments

We would like to thank Sankini de Silva, Ling-Fei Chen and Anju George for participating in data checking processes.

## Author Contributions

**Conceptualization:** Rebecca Featherston, Laura E. Downie, Adam P. Vogel, Karyn L. Galvin.

**Data curation:** Rebecca Featherston.

**Funding acquisition:** Laura E. Downie, Karyn L. Galvin.

**Investigation:** Rebecca Featherston, Karyn L. Galvin.

**Methodology:** Rebecca Featherston, Laura E. Downie, Karyn L. Galvin.

**Project administration:** Rebecca Featherston.

**Supervision:** Laura E. Downie, Adam P. Vogel, Karyn L. Galvin.

**Validation:** Karyn L. Galvin.

**Visualization:** Rebecca Featherston, Adam P. Vogel.

**Writing – original draft:** Rebecca Featherston, Karyn L. Galvin.

**Writing – review & editing:** Rebecca Featherston, Laura E. Downie, Adam P. Vogel, Karyn L. Galvin.

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
