## [Decision Letter · Decision Letter 0]

18 Jun 2020

PONE-D-19-31005

Decision making biases in the allied health professions: A systematic scoping review

PLOS ONE

Dear Dr. Featherston,

Thank you for submitting your manuscript to PLOS ONE. After careful consideration, we feel that it has merit but does not fully meet PLOS ONE’s publication criteria as it currently stands. Therefore, we invite you to submit a revised version of the manuscript that addresses the points raised during the review process. The reviewers have provided quite detailed and I would recommend you carefully consider these and respond accordingly. During the revision process, I would suggest you also consider adding and/or revising the key take home messages from the manuscript (the "so what").  

We look forward to receiving your revised manuscript.

Kind regards,

Saravana Kumar

Academic Editor

PLOS ONE

Journal Requirements:

No authors have competing interests. 

We note that one or more of the authors are employed by a commercial company: Redenlab.

Reviewers' comments:

Reviewer's Responses to Questions

**Comments to the Author**

1. Is the manuscript technically sound, and do the data support the conclusions?

Reviewer #1: Yes

Reviewer #2: Yes

2. Has the statistical analysis been performed appropriately and rigorously? 

Reviewer #1: N/A

Reviewer #2: Yes

3. Have the authors made all data underlying the findings in their manuscript fully available?

Reviewer #1: Yes

Reviewer #2: Yes

4. Is the manuscript presented in an intelligible fashion and written in standard English?

Reviewer #1: Yes

Reviewer #2: Yes

5. Review Comments to the Author

Reviewer #1: Thank you for the opportunity to review this paper which (as someone with a technical/research interest in the area) was informative in outlining the scope of the literature regarding AHPs and cognitive (and other) biases.

The review was very well written, it was readable, and the scoping review process undertaken diligently - I have no comments to make on the presentation, structure or science behind the paper.

I do however have one major issue with the paper in its current form and some minor points that I think – if addressed – could improve the manuscript.

Taking the major issue first; you have done a good job of describing the who, what and where the biases are in AHPs but you have provided no sense of “so what”? In other words, these biases exist but I (the reader) am none the wiser as to whether they actually matter or not. For example, if I routinely and systematically deny possibly effective therapy [which themselves are subject to considerable uncertainties in effectiveness) to 1:50 of my aboriginal patients is this of the same order of magnitude (clinically, not morally, compared to my colleague who also systematically spends more time and is more conscientious with all their white patients. These are perhaps reductio ad absurdum examples, but hopefully you see my point. There are biases and BIASES… some sense/summary and synthesis of what the effects of these biases are on clinical processes and outcomes, decision success criteria would significantly enhance the sense of “well, that paper mattered” in the reader.

I realise that the studies will be very heterogeneous so I am not suggesting some kind of meta-analysis in the statistical sense, just a guided tour and interpretation of what the effects of these biases on decision quality, outcomes, variations in practice could be.

Comparatively minor issues I would like to see addressed include:

1) p4 110-112: I concur with your approach to pulling together a range of biases using systematically altering a decision outcome. However, biases generally come with negative connotations: they systematically alter a decision outcome from some prescriptive/normative idea of “true”/"good"/"quality" (either correspondence - with some empirical criterion or coherence - against some theoretical/logical/probabilistic criterion. if a "bias" systematically alters a decision towards "better" then this isn’t really a problem is it? The negative impact of biases is implicit in your text, but should be made explicit.

2) p6 148 – 150: would like to see an assessment of the possible effects of biases on decision process and outcomes in AHPs as an aim (see above)

3) p6 169-172: although AHPs as part of a team could be problematic as "decision making" rarely a solo exercise. The MDT itself likely to introduce different decision processes/reasoning.... c.f. Kenneth Hammond’s idea of the cognitive continuum and the ways in which "visibility" of judgements (i.e. having to explain your rationale/reasoning to MDT colleagues) impacting on reasoning and cue use (and presumably biases). (Hammond, K.R., Human judgement and social policy : irreducible uncertainty, inevitable error, unavoidable injustice. 1996, New York ; Oxford: Oxford University Press.). Perhaps worth flagging in the discussion?

4) p7 179-183: very medical range of uncertainties/decision types - which is OK - BUT misses the possibility that AHPs may make additional decisions that lie outwith this typology: c.f. Thompson and nurses and the addition of communication and existential decision types perhaps something to flag as a possible omission/limitation in the discussion https://ebn.bmj.com/content/7/3/68

5) p7 outcome measures: how did you control for selection (bias) in studies being reviewed in which the two different groups seeing two different sets of vignettes were not randomly assigned? the study DESIGNS being reviewed could themselves have introduced biases (not cognitive) but impacting on the interpretation or estimation of effects?

6) study designs (205-210): RCTs and pre post designs with a control are not the same (in terms of trustworthiness of the results observed... they are of course not invalid but do need to be intellectually adjusted for in the narrative synthesis: ie. higher quality study designs demonstrate lower effects perhaps? Again, needs flagging in discussion for the reader (or detail in the methods if done)

Overall it’s a thoroughly competent piece but I think you could widen its appeal and interest even further.

Reviewer #2: Concrete Suggestions:

Line 70: “A lack of consistency in this process can have serious ramifications for patients, clients, and clinicians.” �Is consistency the right word? May not require consistency since clinical decision making should cater to the contexts of each individual patient. Nitpicky (and not an error), but would suggest picking a different word. Lack of consistency is arguably necessary.

Line 72: Are any cognitive reasoning processes necessarily inherent? Unclear to me what the distinction between hard wired or learned reasoning processes is, and why they are meaningful.

Define allied healthcare professionals earlier in the paper.

Would add a paragraph in the introduction of the paper rationalizing why it’s important to also consider the effect of biases on decisions/outcomes by allied health professionals explicitly. Why the focus on allied health professionals? How are they distinct (or not) from physicians, who seem to be a robust subject of study? Why are they different (or not) from accountants, bankers, loan managers, teachers, cafeteria workers, waiters, screenwriters who all also presumably have sets of biases that affect their work? (I think it would be helpful for the authors to reason their scope of investigation.)

I think it would also help if they provided concrete examples of how biases in the allied health professions cause damage/harm/difference in outcome. The reader can generate some through their own imagination, but I think having explicit examples to ground the paper would make it more immediately powerful/applicable/interesting/compelling to readership.

Overall Impression:

This paper meets scientifically rigor and does an excellent job composing a descripting summary of existing research on biases pertaining to allied health professionals. It does give a general view and broad survey, but it also becomes general to the point that the applicability is diluted. Though the paper is informational and does include an explanation of biases and their history, but it doesn’t target or apply to a specific problem. At the end, it reads very broadly, like “there are many biases and they can do many things across outcomes pertaining to allied health professionals,” which is not a particularly compelling conclusion.

As a report on the state of current bias research on allied health professionals, it does the job well. But it’s not particularly satisfying in terms of application, next steps, or what this means for allied health professionals, researchers, or patients—besides the fact that biases are present and they are sometimes studied, though sometimes studied not that well.

The conclusion itself say that it reviews studies that investigate biases, and that further research is needed—especially those that investigate real-life healthcare decision making (instead of using vignettes in the methodology). So while the manuscript perhaps achieves what it sets out to do, it is not a particularly compelling piece of scholarship. Of course, there is a role for research that lays out the state of the state without identifying clear patterns or advocating for specific actions. So, I think it is a clearly designed paper, and well-executed for the purpose that it seeks to achieve (a review looking at what kind of investigations there are regarding biases that affect allied health professionals) but it’s not all that applicable or compelling.

The authors provide many observations. But, without synthesizing strong conclusions from the available literature/existing body of scholarship, it is uncelar what impact this paper would have on addressing clinical biases on the practice of allied health professionals. I do completely agree that the investigation of biases is extremely important, but this manuscript is more of a report than a furthering piece of scholarship, though I recognize that I have my own biases in expecting that scientific research take some stance or advocate for something concrete.

Overall, I think this paper would benefit from a more concrete synthesis instead of spending a bulk of its content on a descriptive summary. Without a stronger conclusion, the reader is left unsure on the take away, which undermines the clarity and purpose.

6. PLOS authors have the option to publish the peer review history of their article (what does this mean?). If published, this will include your full peer review and any attached files.

Reviewer #1: No

Reviewer #2: No

---

## [Author Response · Author response to Decision Letter 0]

31 Jul 2020

Please find our responses to each of the reviewer comments in the document 'Response to Reviewers'.

---

## [Decision Letter · Decision Letter 1]

2 Oct 2020

Decision making biases in the allied health professions: A systematic scoping review

PONE-D-19-31005R1

Dear Dr. Featherston,

We’re pleased to inform you that your manuscript has been judged scientifically suitable for publication and will be formally accepted for publication once it meets all outstanding technical requirements.

Kind regards,

Saravana Kumar

Academic Editor

PLOS ONE

**Comments to the Author**

1. If the authors have adequately addressed your comments raised in a previous round of review and you feel that this manuscript is now acceptable for publication, you may indicate that here to bypass the “Comments to the Author” section, enter your conflict of interest statement in the “Confidential to Editor” section, and submit your "Accept" recommendation.

Reviewer #1: All comments have been addressed

2. Is the manuscript technically sound, and do the data support the conclusions?

Reviewer #1: Yes

3. Has the statistical analysis been performed appropriately and rigorously? 

Reviewer #1: N/A

4. Have the authors made all data underlying the findings in their manuscript fully available?

Reviewer #1: Yes

5. Is the manuscript presented in an intelligible fashion and written in standard English?

Reviewer #1: Yes

6. Review Comments to the Author

Reviewer #1: Happy that previous review comments addressed. I think the ultimate "real world" value of this paper is in framing real life examples provided by clinicians that illustrate the real world implications of biases in fostering and sustaining inequalities (in access) and unwarranted variations in practice. Would be worth developing this as part of the dissemination strategy for the paper - encourage critical debate. Well done on summarising this for the first time.

7. PLOS authors have the option to publish the peer review history of their article (what does this mean?). If published, this will include your full peer review and any attached files.

Reviewer #1: **Yes: **Professor Carl A Thompson

---

## [Editor Report · Acceptance letter]

7 Oct 2020

PONE-D-19-31005R1 

Decision making biases in the allied health professions: A systematic scoping review 

Dear Dr. Featherston:

I'm pleased to inform you that your manuscript has been deemed suitable for publication in PLOS ONE. Congratulations! Your manuscript is now with our production department. 

Kind regards, 

on behalf of

Dr. Saravana Kumar 

Academic Editor

PLOS ONE